# National School Feeding Program (PNAE): A Public Policy That Promotes a Learning Framework and a More Sustainable Food System in Rio Grande do Sul, Brazil

**DOI:** 10.3390/foods12193622

**Published:** 2023-09-29

**Authors:** Eliane Alves da Silva, Eugenio Avila Pedrozo, Tania Nunes da Silva

**Affiliations:** School of Administration, Federal University of Rio Grande do Sul, Porto Alegre 90010-460, Brazil; eliane.alves@ufrgs.com.br (E.A.d.S.); eugenio.pedrozo@ufrgs.br (E.A.P.)

**Keywords:** learning, political economy, PNAE, public policy, food system

## Abstract

Food systems drive change, which can accelerate the process of ending hunger, improving diets and protecting the environment. This is the attribution given to the Brazilian National School Feeding Program (Programa Nacional de Alimentação Escolar—PNAE), which was created to supply 15% of the food needs of millions of children in Brazilian schools. Therefore, the purpose of this article was to create a framework to analyze learnings in the PNAE that contribute to the development of a sustainable food system in the state of Rio Grande do Sul, Brazil. For this, a qualitative descriptive survey with abductive logic was chosen through a study of multiple cases and semistructured interviews as its strategy. It was noticed that individuals tended to group themselves in accordance with their learning level. A group in which instrumental learning was prevailing would tend to strictly follow the rules. Groups that already had communicative learning would be more proactive and look for improvements. However, when they would reach emancipatory learning, they would become more willing to disrupt initiatives to create new perspectives to solve problems. From that, decisions became political, and the more organized the groups became, the more power they had to allow their ideas to prevail.

## 1. Introduction

With the development of territorial approaches and new forms of governance, the involvement of different stakeholders in the production, processing, distribution and consumption of food in a given territory has begun to be considered, including not only farmers and economic agents, but also technical assistance, rural extension, research, public policies, consumers and organized civil society [1]. This organization of production and consumption is known as the food system.

According to Von Braun et al. [2], food systems reach different scales, global, regional, national and local, with the possibility of being classified as food systems, provided that they meet two essential criteria: (1) they must be suitable for the purpose in question to support global and national collectives striving to promote positive change, while accelerating the process to end hunger, improving diets and protecting the environment; and (2) they must define domains for policy and programmatic priorities so as not to exclude all economic, social and ecological dimensions of sustainability.

Growing enough nutritious food for everyone requires healthy, nutrient-rich food, soils, abundant fresh water, biodiverse crops and a stable climate [3,4,5]. This idea of a systemic balance, however, has not yet been accepted by all productive systems—there are approximately a billion hungry people on the planet—but hunger is caused by poverty and inequality, not by scarcity due to the lack of production. The world already produces enough food to feed nine to ten billion people, with a population peak expected by 2050. Solutions to hunger and food supply must consider the distribution of food and access to income, land, seeds and other resources [6].

In this context, it is important for the state to formulate and strengthen public policies that guarantee changes in dietary patterns and the rearrangement of the food system, in addition to promoting agricultural production, guaranteeing access to healthy foods and enabling fair trade and sustainable rural development. This articulation dialogues with the precepts of intersectorality and interinstitutionality, which are inherent to the success of policies guided by the norms of sovereignty and food and nutritional security (SSAN) [7].

Given this, other tools and commitments are required to nurture more sustainable and equitable food systems. As pointed out by Duncan, Levkoe and Moragues-Faus [8], a number of political economists have acknowledged the importance of hybrid approaches advocated in cultural and poststructural geography to understand the dynamics of the food system and expand to other thematic areas beyond food production.

A starting point for this analysis are the four questions proposed by Bernstein [9], which would be: Who owns what? Who does what? Who gets what? What do they do with that? The author highlighted the analytical usefulness of these questions, which could be applied from individual farming families to villages, local and national socioeconomic units of investigation and global economics. Furthermore, these questions serve to identify the main power relations in relation to social interactions and the ways in which they impact and influence decision making in food systems.

With that, it becomes clear that each food system is the result of social learning, which provides resource management perspectives, considering the social, cultural and political spheres [10]. The modified understanding of evolving conditions and, consequently, adaptations in managed resource systems depends essentially on four questions: Who learns? What is learned? How is it learned? Why do they learn? [11].

Lankester [12], in her study, showed that organized collective learning, adversity and experimentation with skills and techniques for managing natural resources can facilitate critical reflection on practices, individual questioning, according to Habermas [13,14], and collective questioning, according to Mezirow [15].

Based on this context, the question arises: How can learning influence the construction of a food system? Evidently, to answer this question, an empirical example is needed. In Brazil, since the 1940s, there has been a concern regarding the eradication of hunger and, especially, combating child malnutrition. In this context, the National School Feeding Program (Plano Nacional de Alimentação Escolar—PNAE) emerged, which aims to meet at least 15% of the food needs of students. The program evolved over the years until the passing, on 16 June 2009, of Act 11,947 [16], which establishes that at least 30% of food purchases for school meals must come from family farmers. Thus, each municipality was able to organize itself, alongside local farmers, to provide food for schools.

These arrangements agreed on between the municipal, state and federal governments, different productive stakeholders, union leaders, teachers, nutrition professionals, students, parents and other institutions involved, in addition to political pressure, influenced the final result of this food system oriented toward the institutional market. Each Brazilian municipality, within what is allowed in the PNAE legislation, establishes its rules and priorities. Additionally, within a state, different ways of producing and purchasing agricultural products for school meals can be found.

The state of Rio Grande do Sul, for example, has an agricultural production that includes crops, such as soy, corn, wheat, rice, beans, tobacco, grapes, apples, citrus and olive trees. This state has 11,088,065 inhabitants [17], of which 2.4 million are students [18]. It comprises 497 municipalities divided into seven mesoregions—northeast, northwest, southeast, southwest, west central, east central and metropolitan areas [17]. In each city, in each mesoregion, it is possible to identify the diversity in the execution of the PNAE. Therefore, the goal of this article was to prepare a framework to analyze the learnings in the PNAE that contribute to the development of a sustainable food system in the state of Rio Grande do Sul.

Research of this nature is important, as it shows not only how public policies influence the construction of food systems, but also how learning, both at the individual and collective level, influences the political economy behind the management of this public policy. When analyzing the questions: “who owns what?”, “who does what?”, “who gets what?”, “what do they do with that?” it is possible to answer the questions “who owns what?”, “who performs what?”, “who receives what?” and “what happens to it?”. Additionally, a better understanding of people reflects an economic and political organization that is better suited to the local food system.

To achieve the proposed objective, a qualitative descriptive survey was chosen [19,20,21] using abductive logic [15,22,23] with a multicase study [24,25] and semistructured interviews [26] as the strategy.

This article is divided into six sections: Section 1 is the introduction; Section 2 addresses the empirical theoretical framework; Section 3 describes the methodological procedures; Section 4 presents the results; Section 5 presents the discussion about the results; and Section 6, at the end, presents the conclusion.

## 2. Theoretical Framework

In this section, theoretical approaches are discussed that should lead to a better understanding of the proposed framework, in addition to a brief description of the National School Feeding Program (PNAE).

### 2.1. Food Systems and Political Economy

According to von Braun et al. [2], to be classified as a food system, a productive system should meet two essential criteria. The first is that it must be suitable for the purpose in question in order to support global and national collectives striving to bring about positive change, accelerating progress towards ending hunger, improving diets and protecting the environment. In turn, the second criterion consists of defining the domains for policy and programmatic priorities so as not to exclude all economic, social and ecological dimensions of sustainability.

Efforts to visually map food systems can help researchers and decision makers to identify key interactions and the natural and social mechanisms that regulate these interactions. Nevertheless, this is a complex analysis, involving economic, social or biophysical factors. Consequently, there is no defined way out of this dilemma, depending largely on the relevant policy issue, within a context and the scale of food systems considered [2].

Food systems involve several stakeholders and their added value interconnects activities related to the production, aggregation, processing, distribution, consumption and disposal of products from agriculture and farming, coupled with a wide variety of economic, social and environmental environments [27,28]. It should be noted that sustainable food systems are those that contribute to the food and nutritional security of all, considering economic, social, cultural and environmental factors.

A political-economy-oriented approach to food systems incorporates a broad historical and geographic perspective, helping to explain why and how power changes over time and how the activities of one group affect others [29].

Bernstein [9] summarized the main concerns of a political economy analysis in four main questions—Who owns what? Who performs what? Who receives what? What happens to it?—highlighting the analytical usefulness of these questions, which can be applied in different places and scales. Additionally, these questions aim to discover the main power relations in relation to social interactions and the ways in which they impact and influence decision making in food systems.

These questions do not serve as a summary of actual historical change and development. Nevertheless, they are useful for organizing a possible investigation. Such an investigation can always reveal a great wealth and diversity of historical realities, forms and patterns, trajectories and results. In this sense, creating history means always identifying and seeking to explain specificity, leading to many other additional determinations [9].

According to Bernstein [9], the agricultural sector is the result of economic interests and their institutions, in addition to specialized practices that affect the activities and reproduction of farmers. According to the author, agriculture refers to the ways in which the conditions of production are secured before agriculture itself can begin, thus, including the provision of work instruments or inputs, understood as tools, fertilizers and seeds, as well as land, labor and credit markets.

Food systems involve building blocks, identifying actions that leverage positive side effects or remedy negative side effects of policies. These side effects, in turn, before generating political results, undergo various stakeholders, which have an individual and collective learning system. The stakeholders of food systems determine their decisions based on their perception of the world, i.e., based on what they have learned, which means that before moving forward with the study of food systems and political economy, it is necessary to understand the learning process of these individuals.

### 2.2. Learning and Food Systems

According to Jürgen Habermas [13,14], learning is seen as an inherently motivational, cognitive, affective–emotional and social process. Its broadly interpretive approach sees sociocultural life constituted by the meaningful action of people, which gives rise to the problem or interpretive understanding. Habermas [13] proposes parallel relationships between the orientations of self-directed learners and the three areas in which human interest produces knowledge. Thus, he describes three major areas in which human interest generates knowledge: the technical, the practical and the emancipatory. They are grounded in relationships with the environment, other people and power, respectively. Each area is “knowledge-constitutive” because it has its own distinct categories for interpreting experience, methods for discovering knowledge and methods for validating claims pertinent to it [15,30].

Habermas views the first two human interests as representing distinct domains of learning, comprising the domains of instrumental and communicative learning, respectively. In turn, the emancipatory interest involves a learning dimension of critical reflection with implications for the other two. Each of the two interests also require a different mode of personal learning, with different learning needs and different implications for adult learning facilitators.

For Mezirow [15], when Habermas [13,14] pointed to the critical social theory as the most appropriate systematic investigation process to study material related to emancipatory human interest, he was referring to critical reflection on cultural assumptions or, more specifically, ideological in the field of communicative learning.

Learning by individuals through observation or interaction with their environment results in the social learning perspective being interesting for sustainable food systems. Many other perspectives tend to focus on individual or structural incentives as determinants of human behavior. When discussing food systems, social learning becomes useful for conveying the way in which people learn.

From a social learning perspective, communicative rationality [13,14] is the guiding principle for such interaction. Through dialogue and deliberation, problems and issues are identified and alternatives explored. Further, based on subsequent shared understanding, decisions and actions can be adjusted if necessary. This, however, does not mean that this process occurs without conflict. Practice has proven that communication can be a source of conflicts, although it also can be a means of resolving them. The ideal of communicative rationality can be a guide in carrying out an interaction in which all parties who feel such a need are free and have an equal chance to express their views, doing so in an understandable, legitimate and truthful way [11,31].

From a normative standpoint, social learning provides an alternative to traditional resource management perspectives, enabling a collective and collaborative learning process that links the biophysical to the social, cultural and political spheres [10]. As such, a social learning perspective can be a guiding framework for carrying out continual adaptation in resource systems. The modified understanding of evolving conditions and, consequently, adaptations in managed resource systems depends essentially on four questions: Who learns? What is learned? How is it learned? Why do they learn? [11].

According to Mezirow [15], there is a need for a theory of learning that is able to explain how adult learners give meaning or significance to their experiences, the nature of the structures that influence the way they build the experience, the dynamics involved in the modification of meanings and the way in which meaning structures themselves suffer and change when students find them dysfunctional. These understandings must be explained in the context of adult development and social goals.

A meaning-centered theory of learning could provide a solid foundation for a philosophy of adult education from which appropriate practices in goal setting, needs assessment, program development, instruction and research could be derived [15]. Lankester [12] advances in the transformative learning theory proposed by Mezirow [15] by linking the level of individual learning to social learning. The author adapted the experiential learning cycle developed by Kolb [32] and adapted by Leeuwis [33] to the transformative learning framework developed by Tarnoczi [34].

Initially, this association between learning, food systems and political economy may seem abstract, but when it is related to public policy, used to reorganize a food system, the relationship between these approaches becomes clearer. The next topic discusses the origins, operationalization and types of management present in the PNAE.

### 2.3. PNAE

According to Peixinho [35], the National School Feeding Program (PNAE) had its origins in the Social Security Food Service (Serviço de Alimentação da Previdência Social—SAPS), which was founded in August 1940. During the 1950s, Josué de Castro, Federal Deputy and President of the Executive Council of the Food and Agriculture Organization of the United Nations (FAO), aimed to raise world awareness of the issue of hunger and extreme poverty, promoting projects that highlighted hunger and its possible solution through action and the will of social stakeholders [35]. Silva [36] mentions that, in 1952, the Food Conjuncture and Nutrition Problems in Brazil plan was prepared, including nutritional issues, the expansion of school meals, food assistance for adolescents, regional programs, enrichment of basic foods and support to the food industry. This project resulted in the School Lunch Campaign (Campanha da Merenda Escolar—CME), established through Decree No. 37,106, instituted on 31 March 1955 by President Getúlio Vargas [37,38].

The period between 1955 and 1970 was characterized by the predominance of participation by international organizations in the PNAE. The 1960s were strongly marked by the presence of food from the United States, financed by the United States Agency for International Development (USAID) and by the World Food Program (WFP) of the United Nations Organizations [35,38].

According to Schwartzman et al. [39], starting in 2003, actions related to food and nutrition security were placed as a priority in the country’s development agenda. During this process, several policies related to the topic were prepared or strengthened. The Fome Zero (“Zero Hunger”) strategy was one such policy, with the PNAE being included as a priority. That same year, the Food Acquisition Program (Programa de Aquisição de Alimentos—PAA) was created through Article 19 of Act 10,696 [40] of 2 July 2003, with the basic purposes of promoting access to food and encouraging family farming. It was based on the PAA that the purchase by family farming was consolidated.

Act 11,947 [16] of 16 June 2009 was the result of an intersectoral process within the federal government and the participation of civil society through the National Council for Food and Nutritional Security (Conselho Nacional de Segurança Alimentar e Nutricional—CONSEA) [41], in addition to efforts at union mobilizations such as Grito da Terra Brasil [42,43,44].

This law universalized the PNAE for all basic education, i.e., from kindergarten to high school, in addition to young people and adults. Defending food and nutrition education as a priority axis, thereby strengthening the community’s participation in the social control of the actions developed by the states, the federal district and municipalities, it provides support for sustainable development, with incentives for the purchase of diverse foodstuffs produced locally, respecting seasonality, culture and food tradition, in addition to prioritizing organic and/or agroecological foods in school meal menus [35].

Following the introduction to the history of the PNAE, it is important to understand how it works and which institutions contribute to its management. The program serves students across all basic education schools (kindergarten, elementary school, high school, and youth and adult education) enrolled in public schools, philanthropic schools and community entities (partnerships with public authorities). The federal government transfers to the states and municipalities and federal schools 10 monthly installments to cover 200 school days, depending on the number of students enrolled [37,38]. Values are standardized and defined according to the levels and modes of education.

From an operational standpoint, the following participate in the PNAE: the federal government, through the FNDE, which is responsible for defining the program’s rules; executing entities, which comprise the education secretariats of the states, the federal district and the municipalities; and federal schools. Executing units are characterized by being a nonprofit civil association established as a legal entity under private law and linked to the school system, which can be instituted through the initiative of a school, the community or both. Such executing units may be called a “school box”, “parent-teacher association” or “parent-teacher circle” [37,42].

The School Meal Council (CAE) is responsible for monitoring the acquisition of products, the quality of food offered to students, the hygienic–sanitary conditions in which food is stored, prepared and served, distribution and consumption, financial execution and the task of evaluating the accountability of the executing entities and issuing a conclusive opinion [42,44].

The Federal Court of Auditors and the Ministry of Transparency, Inspection and Comptroller General of the Federal Government are government bodies that also supervise the program. The Federal Public Prosecutor’s Office, alongside the FNDE, receives and investigates allegations of program mismanagement. Health and agriculture secretariats of the states, the federal district and the municipalities can collaborate with the PNAE through sanitary inspections by certifying the quality of the products used in the food offered and by articulating the production of family farming. The CAE is responsible for carrying out inspections at schools and education secretariats [42,44].

Executing entities have the autonomy to define their way of managing PNAE resources. In centralized management, the financial resources of the FNDE are transferred to a bank account managed by the executing entity. It is responsible for carrying out the bidding process and the public call for tenders, in addition to purchasing foodstuffs, which are supplied to school units for the preparation and distribution of school meals. The delivery of foodstuffs by suppliers can be carried out directly to school units, and there may be central warehouses for the intermediation of supplies [45].

In the case of decentralized or school-based management, the executing entity transfers financial resources to executing units—the school units—which directly purchase foodstuffs for the preparation and distribution of school meals. The program’s financial resources are transferred by the FNDE through the PNAE card account. In this type of situation, the executing entity may carry out the bidding process and the public call. Nevertheless, it is the responsibility of the school to sign contracts for the purchase of foodstuffs and payment using the magnetic card provided by the executing entity to the school [45].

To ensure decentralized management, the executing entity must ensure the necessary structure for: (1) carrying out the proper bidding process and/or acquisition of foodstuffs from family farming; (2) the ordering of expenses, management and the execution of administrative contracts; (3) inventory control and storage of foodstuffs; and (4) rendering of accounts and other acts related to the correct use of financial resources. It should be noted that the school unit must have a school council, which should be registered in the National Register of Legal Entities (Cadastro Nacional de Pessoa Jurídica—CNPJ) and be subdivided into commissions to manage resources, purchase foodstuffs and receive and check products [45].

In semidecentralized or partially schooled management, the executing entity combines centralized and decentralized forms of management. In general, what determines whether a school unit manages resources is the fact that it has a school board. In the event that the school does not have a school board, the executing entity manages FNDE resources [45].

It is also possible for philanthropic entities, community schools and religious schools to receive financial resources for school meals. Such resources are transferred to the respective municipality or state, which is required to serve them by providing foodstuffs and/or transferring the corresponding financial resources. The transfer of financial resources depends on the configuration adopted and on the centralized or decentralized management [45].

The executing entity may choose to purchase meals through outsourcing services. Nevertheless, it would only be able to use the resources transferred by the FNDE to the PNAE account for the payment of foodstuffs, leaving the other expenses necessary for the provision of these meals at its expense, to be covered with its own resources. In this case, the executing entity must hold separate tenders—one for the acquisition of goods and another for services. Expenses incurred with PNAE funds must be supported by original or equivalent tax documents, in accordance with the legislation to which the executing entity is bound. In this type of management, the documents must be issued in the name of the executing entity and identified with the name of the FNDE and the program. It is up to the executing entity to implement and maintain an inventory control system for foodstuffs purchased with PNAE resources in order to record all incoming and outgoing goods, provide the up-to-date status of the physical inventory and enable periodic surveys of the amounts received and distributed in schools [45].

It has been observed that the PNAE is characterized by participatory processes at the local level. Participants are exposed to a careful reflection on the implications of the PNAE in their municipalities and based on their learning, at both the individual and social level, they organize themselves to determine decisions regarding the program. Such decisions affect both political and economic aspects.

### 2.4. Analysis Framework

The aim of this article was to analyze the lessons learned in the PNAE that contribute to the development of a sustainable food system in the state of Rio Grande do Sul, Brazil. Thus, with regard to creating an analysis framework, the three perspectives—food systems, political economy and learning—were useful to ensure the partial achievement of this goal.

Food systems, as mentioned earlier, must support collective organizations and promote positive change, accelerating progress towards ending hunger, improving diets and protecting the environment, in addition to defining the domains for public policies that prioritize the economic, social and ecological dimensions of sustainability. Nevertheless, the result of interpreting a food system depends on the worldview of its stakeholders.

In order to understand the structures of the world of life and levels of learning or unlearning processes that correspond to them, one must initially seek to understand people’s actions in terms of the meaning they have for them. This requires the investigator to adopt the performative attitude of a communicative participant, in which both stakeholder and performer belong to the same “universe of discourse” [30]. From a social learning standpoint, communicative rationality [13,14] is the guiding principle for such an interaction. Through dialogue and deliberation, problems and issues are identified, and alternatives are explored, and based on the subsequent shared understanding, decisions and actions can be adjusted, as necessary.

These decisions and actions influence the analysis of political economy, which, according to Bernstein [9], can be summarized in four main questions: Who owns what? Who does what? Who gets what? What do they do with that? These questions can be useful to critique socioeconomic and political dynamics, exposing how power at multiple scales impacts lived experiences and reproduces inequalities and injustices.

A hybrid approach between the relationship of learning levels, based on Lankester [12], and political economy, based on Bernstein [9], can show the connection of adult education and public policy. The way decision makers decide is a reflection of their values and culture. In this case, the kind of food system, sustainable or not, is a result from this. Therefore, to diagnose problems in public management, or even in private sectors, it is necessary to understand this dynamics. Additionally, the framework seen in Figure 1 summarizes this idea.

This framework aims to ensure that individual learning and individual observations of experiences interfere in their relationship with other individuals (instrumental, communicative or emancipatory learning). People who share the same worldview (Who learns?) tend to form groups, organize themselves to achieve certain objectives (What is learned?), and each stakeholder has a motivation that can be the same as the others or different (Why do they learn?). Collective learning can occur in different ways (How is it learned?). These objectives result in a learning product (What is learned?). When the group learns something, it results in a critical reflection that allows answering questions that characterize political economy, leading the social stakeholders to reflect on how they should contribute to this process (Who owns what? Who does what?), who would be the beneficiaries (Who gets what?) and what would be the end result of this organization (What do they do with that?). By answering these questions, it is possible to organize a sustainable food system, which, in the case of this study, was the institutional market promoted by the PNAE. It, thus, promotes an interaction of several levels in a spiral process in continuous development.

## 3. Materials and Methods

In order to achieve the proposed objective, qualitative descriptive research was selected [19,20,21] with abductive logic [15,22,23] using a multicase study [24,25] and semistructured interviews [26] as the strategy. According to Hedrick, Bickman and Rog (1993), to determine the methodological strategy of a piece of research, it is necessary to understand its research question. The authors propose a basic categorization scheme in the form of questions such as “who”, “what”, “where”, “how” and “why”. For Yin [25], the “how” and “why” questions are more explanatory and probably lead to the use of a case study. Thus, this article used a case study, and its research question was “How can learning influence the construction of a food system?”.

It is important to say that the research question, in itself, does not define a case study, but this research presented a triangulation between multiple sources of evidence, comprising a method that encompasses a document analysis, interviews and observations [25]. In this multicase study, a direct comparison was actualized between different examples to achieve the objective of preparing a framework to analyze the learnings from the PNAE, which contributed to the development of a sustainable food system in the state of Rio Grande do Sul.

The chosen cities were part of the survey “Programa Nacional de Alimentação escolar (PNAE) gaúcho: um estudo avaliativo em busca da aprendizagem para o desenvolvimento sustentável do Rio Grande do Sul” (“National School Feeding Program (PNAE) in Rio Grande do Sul: An evaluative study in search of learning for the sustainable development of Rio Grande do Sul”), developed from December 2019 to September 2023. The cities of Erechim, Santa Maria, São Gabriel, Rio Pardo, Pelotas, Bento Gonçalves and Viamão, cities representing the 7 mesoregions of the State of Rio Grande do Sul, were selected. The choice of the cities was due to their performance in implementing Act 11,947. After checking possible cities, a sample was selected according to ease of access, characterizing a nonprobabilistic convenience sample [46].

Interviewees were selected according to Silva et al. [47,48], who identified some key actors as: nutritionists; representatives of the School Food Council (CAE); representatives of rural extension institutions; representatives in agriculture departments; farmers; associations; cooperatives; regulatory entities; and many other actors. Each municipality had a specific configuration, according to its context. However, the initial contact was always with the education departments and contacting nutritionists, which could help in preparing a mapping of potential interviewees. In total, 94 people were interviewed, totaling 53 h, 21 min and 13 s of interview recordings, in addition to 35 h and 40 min of observations. The characteristics of the interviewees and the dates on which the interviews took place can be seen in Appendix A.

It should be noted that there were language adaptations in the approaches used, and some questions were simplified based on the context of the interviewees. The interviews, as well as the observation process, were documented through the use of field notes [47,48]. After transcribing the interviews, a content analysis was used to treat the collected data, organized around three chronological poles: preanalysis, material exploration and the treatment of results, inference and interpretation [49].

## 4. Results

The state of Rio Grande do Sul has 11,088,065 inhabitants [17], of which 2.4 million are students [18]. The state is composed of 497 municipalities divided into seven mesoregions: northeast, northwest, southeast, southwest, west central, east central and the metropolitan regions [17].

In the following sections, descriptions were provided for each municipality visited.

### 4.1. Municipality of Erechim—Northwest Mesoregion

The municipality of Erechim is located in the northwest mesoregion, with 429,164 km of land area and 105,705 inhabitants [17]. Its economic base focuses on agriculture, livestock industry and services. Nevertheless, the industrial sector has greater representation, with approximately 37.53% [17]. The primary sector currently accounts for 6.39% of municipal revenue, and the city has approximately 2520 small producers. They produce, in essence, soybeans, corn, wheat, beans, barley and fruits, and raise poultry, cattle and pigs. The size of the properties is also considerably small, according to estimates, with 95% of the cultivation sites in the region not having an area greater than 100 hectares. The plants with the largest hectare area are, respectively, corn, soy, wheat, barley and beans [50].

The total number of students in the municipal public network is 7500 students, distributed across 18 schools, 7 of which are elementary schools, 1 fine arts school; 9 kindergarten schools, 1 youth and adult education center (Centro de Educação de Jovens e Adultos—CEJA) and the municipal technological center (Núcleo Tecnológico Municipal—NTM).

The school meal department team comprises four nutritionists, one manager from the administrative area and one intern in the nutrition sector. Regarding the number of cooks, the schools have a team of 88, of which 77 are active.

The management of the PNAE is centralized, with the nutritionists being responsible for organizing the bidding processes and the public call for tenders. The resources of the PNAE are used in the public call, including only products from family farming. The municipality contributes with the supplement through the purchase of products acquired through bidding for school meals. There is an effort to purchase all products from family farming, and nutritionists have encouraged cooperatives to create agroindustries to increase the supply of products. Despite its positive effect, this initiative partially undermines the goal of purchasing, with PNAE resources, 100% products from family farming. Due to Article 21 of the FNDE Resolution No. 6 [45], the purchase of school meal items must contain 75% of natural products or, minimally, 20% processed and ultraprocessed foods and 5% of processed ingredients for culinary use. Products such as bread, cakes or flour, for example, would be included in the 20% quota and the quantity to be purchased would be reduced.

The municipality of Erechim created the option to purchase other family farming items that are not purchased through the public call for tenders.

Thus, in addition to investing more than 90% of the funds received from the FNDE in family farming, it also purchases organic and agroecological products at an overpriced value, as established by Act 11,947 [16]. Moreover, it should be noted that this region is marked by the presence of nongovernmental organizations that encourage agroecological production, which is the case of the Center for the Support and Promotion of Agroecology (Centro de Apoio e Promoção da Agroecologia—CAPA) and the Center for Popular Alternative Technologies (Centro de Tecnologias Alternativas Populares—CETAP).

The municipality is served by three cooperatives whose foundation is linked to the PNAE. The CooperFamília cooperative was founded in 2005, with 20 farmers serving the PNAE since 2010. Currently, it has 700 members, who also operate in organic production. Cooperativa Nossa Terra has approximately 3000 family farmers, 18 associated cooperatives and 50 agroindustries, operating throughout the Brazilian territory with a focus on serving institutional programs, such as the PNAE and the Food Acquisition Program (PAA). The Central Cooperative for Family Agriculture Trading in the Solidarity Economy (Cooperativa Central de Comercialização da Agricultura Familiar de Economia Solidária—CECAFES) was established in 2012, with an initial formation of 11 members and currently having 19 associated cooperatives from family farming. This cooperative has its own headquarters, facilities and equipment, with the objective of marketing and entering into potential markets in the region, as well as large consumer centers.

### 4.2. Municipality of Santa Maria—West Central Mesoregion

Santa Maria is a municipality that, according to estimates by the Brazilian Institute of Geography and Statistics [17], has 271,633 inhabitants, distributed across a territorial area of 1,780,194 square kilometers. Its main economic activity is focused on service provision, mainly state and federal public services [51]. Tertiary urban functions absorb over 80% of the active population of the city, comprising mainly the sector engaged in commercial and educational activities. The agricultural sector runs second, followed by the secondary sector, which is characterized by small- and medium-sized industries, focusing on the processing of agricultural products, metallurgy, furniture, footwear and dairy products [17].

The municipality has, on average, 157 educational institutions, including 3 technical colleges, 8 vocational technical education institutions and 2 military colleges. Additionally, it features the Southern Regional Center for Space Research of the National Institute for Space Research, linked to the Ministry of Science and Technology [52].

The Municipal Public Network has a total of 20,000 students, distributed in 80 schools, of which 11 are philanthropic entities. The School Meal Department team is composed of one nutritionist and one employee in charge of accountability. The management of PNAE resources became mixed (schooled) through the Municipal Act 4997 of 27 April 2007 [53]. This type of management comprises a monthly transfer of resources to the school councils of the legally constituted municipal schools. In the event that the schools do not have school councils, school meals are given through a centralized purchasing process under the responsibility of the SMED.

Although each school receives its resources directly, the municipality retains 40% of the amount sent by the FNDE for the purchase of family farming. The purchase of family farming products is centralized through a universal public tender. The cooperative that wins the tender is responsible for delivering the items directly to schools, as there is no central warehouse. Generally, two public calls are performed every year.

The rest of the value—60% of the FNDE funds—is directed to schools, which purchase the remainder of the lunch items by making a small process of three quotes for each item purchased. There is no tender for the purchase of the other foodstuffs—principles buy the goods in local markets.

The municipality is served by three agricultural cooperatives. Nevertheless, only one comes from the region—the Coopercedro cooperative—being part of a federation of cooperatives known as Unicentral. Coopercedro was founded on 30 October 2006 to serve the Food Acquisition Program (PAA). In June 2010, they began to provide services for the municipality’s PNAE. On 18 November 2014, Unicentral was founded, with the aim of participating in more institutional policies. Local cooperatives did not have the capacity to supply all the products in public calls and ended up losing sales to cooperatives in other regions. Hence, to address this problem, Unicentral was created, comprising an association of cooperatives distributed across the region and the state, which, being located in the municipality, has priority in purchases. In turn, products that are not available in the region are obtained from other cooperatives. Another relevant point is that Unicentral works in partnership with CECAFES, a cooperative based in the city of Erechim.

### 4.3. Municipality of São Gabriel—Southwest Mesoregion

São Gabriel is a municipality extending over 5053.46 square kilometers, with 58,487 inhabitants. Its urbanized area is 18.31 square kilometers, and its economic base is agriculture, livestock, fish farming and beekeeping [17].

The total number of students in the municipal public network is 7000 students, distributed across 38 schools, of which 8 are rural schools. The management of PNAE resources has been outsourced since 2009, and since the approval of Act 11,947 [16], the municipality has employed the services of Qualiti Indústria Comércio Serviços de Comida Ltd.a.

The school meal sector, located at the Municipal Secretariat of Education (Secretaria Municipal de Educação—SMED), comprises one nutritionist (technical manager) and one intern. The contract signed with the outsourced company, Qualiti Indústria Comércio Serviços de Comida Ltd.a., establishes that the company should perform continuous food supply services, including the prepreparation, preparation and distribution of meals, with the supply of all foodstuffs and other inputs necessary, logistics, supervision, preventive and corrective maintenance of the equipment and utensils used and cleaning and conservation of the areas covered.

With regard to foodstuffs from family farming, the Secretariat of Education organizes the public call, and qualified farmers deliver their products to Qualiti. The amounts spent on family farming are deducted from the contract, i.e., the municipality pays the farmers directly using PNAE funds.

It should be mentioned that Qualiti serves three departments: the Municipal Secretariat of Education, the Municipal Secretariat of Health and the Municipal Secretariat of Social Assistance. The company’s team consists of two nutritionists and one administrative manager. The nutritionists are divided as follows: they visit the schools, checking the menu, inventory, organization, cleaning and number of meals served, in addition to interacting with the cooks hired by Qualiti. The administrative manager takes care of purchases, managing the central inventory, in addition to interacting with the Secretariat of Education, the RT nutritionist and the financial administrative department.

Despite presenting a counterpart four times greater than the value sent by the FNDE, the municipality of São Gabriel has not been able to reach the purchase percentage of 30% from family farming. There is only one dairy cooperative, Cooperativa Santa Clara, which serves the municipality, while other items in the public call are provided by individual farmers. The municipality was improving its numbers in relation to family farming in 2019, but, due to the pandemic, this number dropped again.

Although the public call notice contains 24 items, only 8 items are actually purchased from family farming. One of the main reasons given by both the nutritionist RT and the Secretary of Education is the lack of interest on the part of farmers in the region to participate in public policies. EMATER is not active in the municipality—at least it does not contact the Secretariat of Education to assist with the PNAE. Another aspect highlighted by the RT nutritionist is the lack of cooperatives in the region, as, according to her, there is resistance regarding this type of organization on the part of farmers. Nevertheless, despite the performance in the purchase of products from family farming, the secretariat intends to resume its efforts to increase its numbers.

### 4.4. Municipality of Rio Pardo—East Central Mesoregion

Rio Pardo belongs to the east central geographic mesoregion. Its area is 2051 square kilometers, and it is located 145 km from Porto Alegre, the state capital [54]. Its economic base is services, agriculture, livestock, industry and public administration [17].

The municipal public network has a total of 2400 students, distributed across 27 schools, 12 Municipal Schools for Early Childhood Education (Escolas Municipais de Ensino Infantil—EMEIs) and 15 Municipal Schools for Elementary Education (Escolas Municipais de Ensino Fundamental—EMEFs). The management of PNAE funds is centralized. The school meal and purchasing sector is composed of 3 people, 1 administrative manager who assists with the school meals, 11 nutritionists and 1 sector manager. Rio Pardo is marked by a peculiarity: in the secretariat, the school meal sector works alongside the purchasing sector.

The municipality of Rio Pardo started to have a greater share of family farming in 2013. It should be noted that the municipality’s contribution is similar to the amount sent by the FNDE. In 2020, with the Coronavirus pandemic, the value declined momentarily—during this period, all students in the network received a food basket.

It is important to emphasize that the region of Rio Pardo was a major producer of tobacco. Therefore, encouraging food production is a way to restructure the municipality’s agricultural vocation, in addition to reducing farmers’ dependence on a volatile market that is not committed to regional development.

### 4.5. Municipality of Pelotas—Southeast Mesoregion

The municipality of Pelotas is located in the southeast mesoregion of the state, with a land area of 1608.78 square kilometers and an estimated number of 325,689 inhabitants. Its urbanized area is 79.39 square kilometers, and it is located 261 km from Porto Alegre, the state capital [17]. Pelotas’ main economic economy is agribusiness and trade, and it is the largest producer of peaches for the canning industry in the country. The municipality is also a major producer of rice and beef cattle, in addition to having the highest milk production in the state (Magalhães, 2011) [55].

The municipal public network has a total of 33,000 students, distributed across 120 schools, of which 100 are urban schools and 20 are rural schools. The management of PNAE resources is centralized. The school meal department encompasses the sector, where the nine nutritionists work the sector for accountability for meals with four servers, the drivers’ room, the management room and the food storage area. Two nutritionists perform more administrative and bureaucratic activities, while another nutritionist works more on the computer, typing the menus into a specific program, analyzing the composition, temperature, cold chambers and 10 freezers. The other six nutritionists work directly with the schools. On average, each takes care of 17 to 19 schools.

The municipality of Pelotas began to have a greater participation in family farming in 2014. The values in family farming follow, in general, a rate ranging between 30 and 48%, reaching 95% in 2017. These advances were thanks to the work of the Municipal Council for Rural Development (Conselho Municipal de Desenvolvimento Rural—CONDER) established by the Secretariat of Education and nutritionists, the Secretariat of Agriculture, the municipal and regional EMATER, the Center for the Support and Promotion of Agroecology (Centro de Apoio e Promoção da Agroecologia—CAPA), cooperatives that serve the municipality and the deputy mayor.

As of 2019, the purchase of items from family farming was increasing. Nevertheless, the Coronavirus pandemic came, and the fact that farming kits were not produced during the year resulted in that number dropping in 2020. According to reports, they even had to return amounts to the FNDE. It should be noted that, during this period, there were no CONDER meetings, so the interactions necessary to guarantee the supply of foodstuffs by cooperatives for the school meals reduced.

In general, a total of three cooperatives serve the municipality, including, especially, Cooperativa Sul Ecológica, which was founded in December 2001 with the aim of helping ecologist farmers in the region to have an organization that represents them in the institutional market promoted by the PNAE.

Another important aspect is that quilombola communities (communities established by descendants of enslaved peoples) in the region supply products to the three cooperatives that serve the municipality. These communities are assisted by CAPA (Pelotas Unit, RS), which also assists cooperatives by forming groups of organic farmers and offering training and technical assistance to these farmers.

A point in common between the municipality of Pelotas and Erechim is the strong influence of nongovernmental organizations that encourage the production of organic and agroecological products. Each location, however, takes advantage of this in a different way.

### 4.6. Municipality of Bento Gonçalves—Northeast Mesoregion

The municipality of Bento Gonçalves is located in the northeast mesoregion, with a land area of 272,287 square kilometers, an estimated population of 123,151 inhabitants and an urbanized area of 34.15 square kilometers [17]. In 2017, there were 10,351 companies operating in all economic categories, with 43% in the service sector, 18% in commerce, 14% self-employed workers, 7% in industry and 18% in other sectors. Additionally, the municipality has 4222 individual microentrepreneurs. Despite that, the industry sector is the one that generates the highest gross revenue for the municipality, reaching the mark of BRL 5.3 billion a year, followed by commerce, with BRL 1.9 billion, with all sectors totaling a gross revenue of BRL 8.62 billion per year to the municipality. Based on that, the participation of sectors in the municipality’s economy is led by the industry, with 65.8%, followed by commerce, with 20.1%, and by the services sector, with 14.1% of the revenue [56].

The municipal public network has approximately 12,875 students, distributed across 45 schools, including 20 kindergarten schools, 2 philanthropic schools, 1 high school and 2 full-time schools established in partnership with the University of Caxias do Sul (UCS). The network has 120 cooks to meet the demand, of which 40 are public officials, while the remainder are outsourced workers.

The school meal department is composed of three nutritionists and one administrative assistant. Relevant aspects were identified, such as Act 3810 of 20 October 2005 [57], which introduced grape juice in school meals. It was reported that the nutritionist job title has existed in the city since the 1990s, long before Act 11,947 [16] determining it mandatory. Another relevant aspect is the fact that the municipality has already won an award for good practices in family farming from the FNDE, as there are projects in the Municipal Department of Education to promote work alongside family farmers. In 2017, they also won the FNDE award.

In the municipality of Bento Gonçalves, since 2012, the participation of family farming has been over 30%, reaching its highest performance in 2019 at 98%. A peculiar feature of the municipality is that it is served by an association of farming groups and individual farmers. This is only possible because the management of the PNAE is centralized. All these farmers are assisted by nutritionists through EMATER, and the results presented show that this has worked very well.

### 4.7. Municipality of Viamão—Metropolitan Region

The municipality of Viamão is located in the metropolitan area and extends over 1,496,506 square kilometers. Its population is 224,116 inhabitants. In 2019, it had an urbanized area of 68.49 square kilometers. Its economic base is in agriculture, livestock, industry and services [17].

The total number of students in the municipal public network is approximately 26,572 students, distributed across 71 schools, of which 13 are kindergartens and 14 are rural schools. There are approximately 125 cooks, of which 25 are public servants and 100 are outsourced workers. It should be noted that, for 2023, 11 children’s schools were expected to be delivered. The department is composed of three nutritionists and one administrative technician. The nutritionists work on the nutritional organization of the program, visiting schools and organizing the menu, while the administrative technician is responsible for taking care of purchases, contacting farmers and organizing public calls and bids.

The municipality is served by six agricultural cooperatives (COMCAVI, COPERAV, COMAFITI, Nossa Terra, CAAF and Ouro do Sul). Another relevant aspect is the Rural Development Council of Viamão (Conselho de Desenvolvimento Rural de Viamão—CONDERV), whose Secretariat of Education has a board to discuss the use of agricultural production in the municipality in school meals, meeting every second Tuesday of each month. During the CONDERV meeting, several relevant aspects were noted—for example, the Municipal Secretariat of Agriculture and Supply is located in the same building as the municipal EMATER. Therefore, there is an interaction between the Secretariat of Agriculture and the five EMATER technicians. CONDERV is a council composed of 10 institutions, including the Secretariat of Education, the Secretariat of the Environment, the Secretariat of Agriculture and Supply, the Secretariat of Social Assistance, EMATER, the Rio Grande do Sul Rice Institute (Instituto Rio Grandense do Arroz—IRGA), a committee dedicated to agrarian rights by the Order of Attorneys of Brazil (Ordem dos Advogados do Brasil—OAB), the rural union, the Cooperative of Agrarian Reform Organic Producers of Viamão (Cooperativa dos Produtores Orgânicos de Reforma Agrária de Viamão—COPERAV), the Mixed Campos de Viamão Cooperative (Cooperativa Mista Campos de Viamão—COMCAVI) and farmers’ associations.

The municipality of Viamão had good numbers in relation to family farming. Although it started in 2013, with a percentage of 19% of purchases of products from family farming, it reached the mark of 94.82% in 2020. In 2021, this number dropped to 29.29%. Until 2020, the municipality did not present a counterpart, but due to the Coronavirus pandemic, in 2021, it invested BRL 12,468,996.29, of which 25% (BRL 3,117,249.07) was directed to kindergartens and philanthropic schools and 75% (BRL 9,351,747.21) to elementary schools and schools for youth and adults. The amount applied to school meals in 2021 was used to purchase food kits for all students in the municipal network.

On 20 December 2022 the school meal department was contacted, as there was an expectation of a counterpart from the municipality. This counterpart took place in September 2022, when the amount of BRL 600,000.00 was invested. It should be noted that the municipality did not provide a counterpart for school meals, but from 2021, there was a greater concern regarding the management in this municipality.

### 4.8. Comparison among the Municipalities Visited

With more in-depth knowledge of each municipality, it was possible to compare their individual level of learning among the organizations and determine some inferences about their organization to attempt Act 11,947 and FNDE resolutions. This can be seen in Table 1.

Municipalities that bet on sectoral councils, such as CONDERV, in Viamão, CONDER in Pelotas or even the farmers’ association of Bento Gonçalves, were those who had communicative learning. This type of learning, because it involves greater collective involvement, drives the development of strategies that not only cover the expansion of items from family farming in the PNAE, but also aims at local development. These municipalities prioritized the purchase of local suppliers.

Emancipatory learning was observed at the CAPA units in Erechim and Pelotas, as well as the Cooperative of Agrarian Reform Organic Producers in Viamão (COPERAV). The three regions visited were those that produced the largest amount of organic food and had an interest in increasing the number of items supplied to school meals. Nevertheless, this was not a simple path. In Erechim, only one cooperative market sold a single organic product to the Secretariat of Education, while the other organic products were resold to municipalities in other municipalities outside the state, such as São Paulo, the state capital of São Paulo, and Curitiba, the state capital of Paraná.

In Pelotas, the two cooperatives served by CAPA, Sulecológica and CAFSul, managed to supply organic and agroecological products for school meals. Both, however, had their own stores to resell their products to other markets. Another relevant aspect was that the quilombola communities in the region produced agroecological foods but were unable to resell the products directly to the Secretariat of Education. They participated as members of the cooperatives or by marketing to them in the cooperatives and reselling their products through them. This in a way harmed the quilombola community, as, according to Act 11,947 [15], they should be given priority in the acquisition of products, receiving a better value for producing agroecological products. What they cannot sell to the municipality is absorbed by local fairs. Thus, there is a need for these quilombola communities to be better supported, so that they can better benefit from public policies.

In the municipalities of Erechim and Pelotas, it was observed that there was potential to acquire organic and agroecological products. Nevertheless, there were bureaucratic obstacles in the way. It was noted that the level of learning of public managers was communicative, while the level of learning of nongovernmental organizations and farmers was emancipatory. This contrast in learning levels did not allow for local development strategies that favored organic and agroecological foods.

Due to it having a decentralized management of resources, the municipality of Santa Maria did not purchase in large quantities, as each school directly received the FNDE resources to buy products for school meals. Nevertheless, the purchase of products from family farming was collectively negotiated, with 40% of the value of FNDE resources being allocated to family farming. It was noted that in relation to public managers and farmers, there was communicative learning, in which participants determined institutional arrangements for local development. In relation to schools and public managers, the level of learning was instrumental, as there was a concern regarding compliance with PNAE standards, although there was no financial contribution from the municipality for schools to buy food; thus, this situation could be characterized as unlearning.

The intercooperativism between cooperatives in Erechim and Santa Maria was highlighted, demoting a communicative learning, which, in this case, promoted regional development. There was an exchange in agricultural products between the northwest and central west mesoregions through CECAFES and UNICENTRAL. In this way, cooperatives became more competitive for public tenders, as they could offer municipalities a variety and greater number of foods.

In the municipality of Rio Pardo, communicative learning was observed, which promoted local development, as the region was initially a major tobacco producer. Therefore, encouraging food production was a way to restructure the municipality’s agricultural vocation, in addition to reducing farmers’ dependence on a volatile market that is not committed to local development. Nevertheless, instrumental learning took place among farmers, as they gathered to participate in the public call, but marketed their products individually, being resistant to structuring a cooperative. In summary, there was not enough confidence to evolve into communicative learning.

São Gabriel opted for the outsourced management of PNAE resources. It was found that the type of learning that prevailed in the municipality was emancipatory, particularly in the relationships between public managers, the contracted company and schools. It was noted that the municipality prioritized the quality of food and there was no restriction on what was invested in to provide this service. As regards to farmers, however, there was only instrumental learning, as no evidence was found that there was an interest in using Act 11,947 for local development. Although there were rural properties in the region, they focused on other markets. In this case, there was also an unlearning, as there was no relationship between the Secretariat of Education and EMATER capable of promoting an increase in the participation of family farming.

## 5. Discussion on Learnings and Political Economy

After visualizing the individual learning level in each municipality including all groups (Who learns?), it was time to discuss collective learning. These groups were created through affinity (What is learned?), people who had similar individual learning and joined with others to start social learning. These individuals organized themselves to determine decisions that were meaningful for them (Why do they learn?). Additionally, this group expanded when it established a way to amplify their learning (How is it learned?).

It was noticed that individuals tended to join themselves in accordance with their learning level. For example, a group where instrumental learning prevailed tended to strictly follow the rules, without questioning or suggestions to improve the process. Groups that already had communicative learning were more proactive and looked for improvements. However, when these people reached emancipatory learning, they were more willing to disrupt initiatives to create new perspectives to solve problems.

From that, decisions became political, and the more organized the groups became, the more power they had for their ideas to prevail (Who owns what? Who does what?). If this process was brought to a public policy, it would be easier to understand. In the case of the PNAE, a group with an instrumental level of learning would establish the beneficiaries in accordance with Act 11,947 and the FNDE resolutions. Groups with communicative learning looked for regional development, promoting diverse sectors while complying with the rules. They used this interaction to obtain more power (What do they do with that?), while organizations whose instrumental learning was its characteristic looked for social improvement and equilibrium among the local powers (What do they do with that?). Regarding emancipatory learning, not all regions reached this level. In fact, only a few organizations in some municipalities reached this stage. Emancipatory learning, in this case, was characterized by the concern to have a sustainable food system, favoring organic and agroecological products. 

However, unlearning could occur. A group could reach the level of communicative learning and regress to instrumental learning. This occurred when the meaning that united them changed, which is what happened in São Gabriel in relation to family farming. Public managers opted for outsourcing services, which solved many problems, but initiatives to encourage family farming were neglected.

When emancipatory learning was achieved, unlearning did not occur. At this level, the group developed critical thinking and, consequently, a new perception of reality. Thus, their personal and social values were consolidated, preventing regression.

Table 2 summarizes this discussion and shows how each type of learning [12] relates to political economy issues, as proposed by Bernstein [9].

In Table 2, it was noticed that each type of learning answered the questions about political economy in a different way. It is worth mentioning that all three types of learning may coexist in a municipality between the relationships of the stakeholders that constitute the food system. The group that has the most power over the others is the one that imposes its will, with the food system being influenced by its type of learning.

It is necessary to understand that different levels of learning in the same food system may cause conflicts; after all, the existing groups have different worldviews and fight for decision-making power over the paths of both food production and consumption. That is why control through public policies is interesting, establishing rules that balance the dispute for power. Nevertheless, it often depends on the level of learning of the dominant group, as the privileges that some groups would have been circumvented. This was observed in relation to quilombola communities and to organic and agroecological farmers. Although this public of farmers was a priority and retained the right to a greater consideration during the purchase, this did not actually occur due to the lack of product certification or even clarifications regarding this priority in the public notice itself.

By analyzing what was learned in the PNAE, it was possible to identify how each municipality visited in the state of Rio Grande do Sul managed the development of a sustainable food system.

## 6. Conclusions

This article sought to reconcile approaches on food systems and political economy with levels of learning, as it is believed that the actions of leaders who dictate the direction of a food system have their origin in how such individuals learn and transmit their knowledge. By understanding how knowledge construction occurs, it is possible to ensure a better understanding of political decisions, which, consequently, influence the economy.

The theoretical framework presented here contributed to the analysis that took in consideration relations of power, since the construction of power through learning processes results from decision makers’ actions. It is also useful for study conflicts, governance problems and public policies, because it identifies the formation of groups and looks to understand their perceptions through learning levels, showing the way that they interpret reality.

As to the PNAE, other policies can be influenced in the same way as between learning levels and political economy. Thus, it is easier to understand some transitions along public policy. For example, the PNAE was influenced in the 1960s by the foods that composed the programs established by the United States Agency for International Development (USAID) and the World Food Program (WFP) of the United Nations. During the 1970s and 1990s, processed foods, such as cookies, chocolate and preserves, were the basis of school meals. It was only in 2009 with the introduction of Act 11,947 [16] that it was determined that at least 30% of food should come from family farming. This measure introduced fresh and healthy food in schools, stimulating the local economy. What really changed was the understanding of program managers from the FNDE to the municipalities that received their resources. Paradigm shifts such as this are due to changes in levels of learning.

The current learning level in each municipality interfered with their performance in implementing the PNAE. Where instrumental learning prevailed, the main interest was to comply with legislation, as there was no search for improvements or innovations. In communicative learning, one seeks to innovate by implementing new processes with the aim of promoting regional development. It is, however, in emancipatory learning that reflection and acceptance of new challenges occur on the part of the stakeholders to act in favor of sustainability and the improvement of society.

The State of Rio Grande do Sul is one of the most developed states in relation to the application of Act 11,947 [16]. Many cooperatives and associations of family farmers have changed the reality of many municipalities. The survey “Programa Nacional de Alimentação escolar (PNAE) gaúcho: um estudo avaliativo em busca da aprendizagem para o desenvolvimento sustentável do Rio Grande do Sul”, financed by the Research Support Foundation of the State of Rio Grande do Sul (Fundação de Amparo à Pesquisa do Estado do Rio Grande do Sul—FAPERGS), contributed to identifying the innovations proposed by each municipality visited. The diversity of results showed the historical and cultural weight of each researched mesoregion. It was expected that the results presented here would help public managers in the implementation or even improvement of the PNAE management, considering that the learning process should come before political decisions.

It should be noted that the research contributed to helping family farmers understand their place in the world and, consequently, their importance in the food production process. By reaffirming their position, they can fight for improvements in public policy that can reflect and improve their reality.

The main difficulty of this research was the limitation in official data on family farming in the PNAE, which had not been updated since 2017. Another limiting factor was the Coronavirus pandemic, which disrupted sectoral councils, as well as relations between the education secretariats with groups of farmers.

Although the framework developed here was applied to the analysis of a food system promoted by a Brazilian public policy, nothing prevents it from being replicated in another country or in other types of food systems. This remains a suggestion for future studies.

## Figures and Tables

**Figure 1 foods-12-03622-f001:**
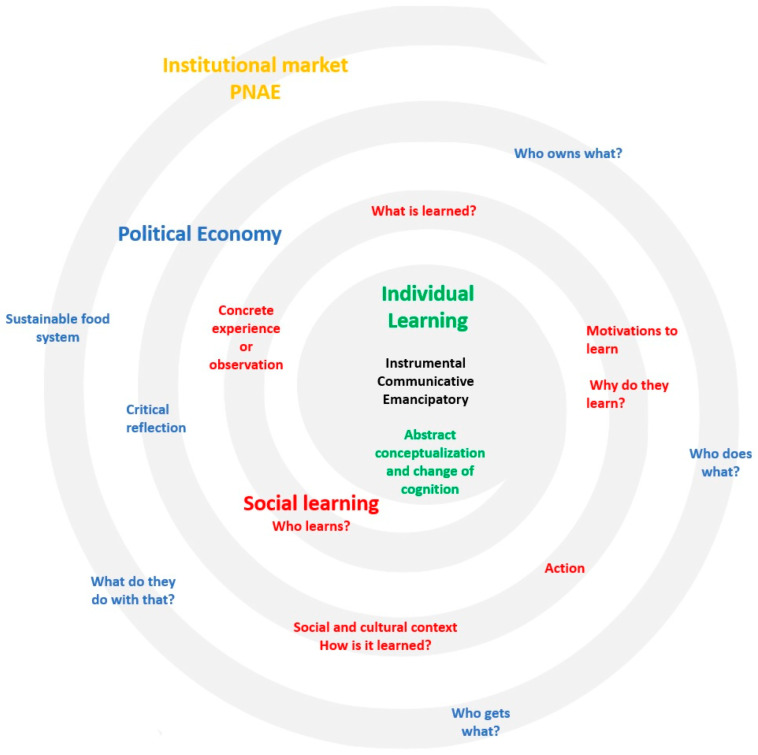
Framework relating to individual and collective learning [12], which, through critical reflection, provides the basis for the analysis of political economy [9], which, in turn, fosters a sustainable food system characterized by the institutional market promoted by the PNAE.

**Table 1 foods-12-03622-t001:** Relationship between individual learning levels in each municipality.

Cities	Erechim	Santa Maria	São Gabriel	Rio Pardo	Pelotas	Bento Gonçalves	Viamão
Financial Resources of the FNDE	Centralized	Decentralized	Outsourcing Services	Centralized	Centralized	Centralized	Centralized
Individual Learning levels							
Instrumental Learning	School managers	Municipal Secretariat of Education	Farmers (there are no cooperatives)	Farmers (there are no cooperatives)	School managers	Farmers’ association	
Communicative Learning	Municipal Secretariat of Education, nutritionists, cooperatives, EMATER, Secretary of Agriculture and Food Security, CAE	Nutritionists, school managers, cooperatives, EMATER, Food Security Secretariat, CAE	Nutritionist, outsourced company, school managers, Municipal Secretariat of Education, CAE	Nutritionist, school managers, Municipal Secretariat of Education, CAE, EMATER	Nutritionist, Municipal Secretariat of Education, CAE, cooperatives, EMATER, EMBRAPA, CONDER, Akotirene, Quilombo	Nutritionists, school managers, Municipal Secretariat of Education, CAE, EMATER, Municipal Secretariat of Health	Nutritionists, school managers, Municipal Secretariat of Education, CAE, EMATER, cooperatives, CONDERV
Emancipatory Learning	CAPA CETAP				CAPA		COPERAV
Unlearning		Municipal Secretariat of Education	Municipal Secretariat of Education EMATER				

Source: Authors’ own work.

**Table 2 foods-12-03622-t002:** Relationship between learning levels and political economy.

Collective/Individual Learning Levels	Instrumental Learning	Communicative Learning	Emancipatory Learning
Who learns?	Nutritionists, public managers, government organizations, farmers, cooperatives	Nutritionists, public managers, farmers, cooperatives, government organizations, union leaders, outsourced companies that provide services for school meals	A few public managers, nongovernmental organizations that promote organic and agroecological traditional communities, cooperatives of organic and agroecological products
Why do they learn?	Act 11,947, FNDE resolutions, regional acts and regulations	Sectorial councils that involve Secretariats, EMATER, farmers, CAE and nutritionists, which adapt Act 11,947 and FNDE resolutions to the municipality’s demand	Need to introduce sustainable consumption, favoring healthier and quality food, in addition to encouraging organic and agroecological farmers
How do they learn?	Following Act 11,947, FNDE resolutions, regional laws and regulations	They meet with representatives of sectoral groups to discuss improvements to Act 11,947 and regional regulations	They meet with public managers and EMATER to introduce organic and agroecological foods in school meals
What is learned?	Following PNAE regulations and complying with Act 11,947 by purchasing 30% of products from family farming	They organize a food system that exceeds the minimum standards required by Act 11,947, promoting the local economy	Demanding means for organic and agroecological products to be introduced into the food system promoted by the PNAE
Political Economy			
Who owns what?	Public managers have power and responsibility for the execution of the PNAE Farmers have their production to meet the PNAE Students have the right to school meals	Public managers participate in the sectoral council and have the power to execute the PNAE and direct the local food system Farmers have incentives to produce more Students have the right to school meals	A few public managers and nongovernmental organizations are aware of the need to introduce organic and agroecological foods in school meals Farmers have their property and organic or agroecological production Traditional communities have their lands and traditions Students have the right to school meals
Who does what?	Nutritionists, public managers and government organizations implement act 11,947 and FNDE resolutions Farmers deliver their produce	Public managers and the sectoral council implement strategies to expand the PNAE and local development Farmers seek to expand their production	A few public managers and nongovernmental organizations seek to create mechanisms to introduce organic and agroecological foods in the PNAE Farmers seek to guarantee organic production conditions
Who gets what?	Public managers receive the maintenance of sending resources Farmers are compensated for their products Students receive school meals	The municipality exceeds the target established by Act 11,947, reaching 100% of products from family farming Farmers are able to expand their production, serving the PNAE and other markets Farmers have production to meet the PNAE, other programs and other markets Students receive a diversified school meal	A few public managers and nongovernmental organizations manage to guide consumption in the municipality towards organic and agroecological foods Organic and agroecological farmers are able to sell their products at a fair price Students receive healthy, pesticide-free school meals
What they do with that?	Public managers execute the PNAE Farmers drive the local economy Students have at least 15% of their dietary needs met	Public managers and the sectoral council foster local development Farmers, with more resources, expand their properties and production and move the local economy Students have at least 15% of their dietary needs met	A few public managers and nongovernmental organizations consolidate the production of organic and agroecological products in the municipality Organic and agroecological farmers obtain certifications Students have at least 15% of their dietary needs met

Source: Authors’ own work.

## Data Availability

The data presented in this study are available on request from the corresponding author.

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
