# Peer review of "National School Feeding Program (PNAE): A Public Policy That Promotes a Learning Framework and a More Sustainable Food System in Rio Grande do Sul, Brazil"

_foods, 2023, doi:10.3390/foods12193622_

Round 1

Reviewer 1 Report

The objective of this article was to create a framework for examining the insights gained from the PNAE, with the aim of fostering the growth of a sustainable food system. The manuscript presents a current and important topic in the field of School Feeding Programs and sustainable food systems. The introduction and the theoretical framework have been thoroughly developed by the authors, but the methodological description and results still need to be refined.

My comments are listed below:

The results and conclusion are missing from the abstract, I recommend restructuring the abstract.

I would suggest expanding the methodological part along the following questions: • On what basis were the interview partners selected? Were there inclusion or exclusion criteria? • Were interviews also conducted with actors from family farms? • What were the interview questions? What were the interview questions based on? It may be worth adding the question list as a supplementary material. • How long did an interview take on average? Where did the interviews take place? • During the analysis of the interviews, did additional subthemes emerge within the main themes?

The results partly show numerical data on the data of individual municipalities, I recommend expanding the results section with the qualitative results of the analysis of the interviews. It might be worth displaying a couple of verbatim quotes from the interview partners.

In the first table, it would be advisable to separate the individual parts in the listings with commas.

Does the first table reflect the results of the interviews?  

Minor editing of English language required.

Author Response

Dear Reviewer 1

We appreciated your comments and suggestions. They contributed for some corrections and changes in our article.

We would like to answer your points showing the main changes that we done.

Best regards

1.      The abstract would lead me too think this is about the development of a tool for gathering data [Therefore, the purpose of this article was to prepare a framework to analyze learnings ], but later on various terms are used with a confusing effect, so aim, purpose etc, see page 9 [The aim of this article is to analyze the lessons learned in the PNAE that contribute to the development of a sustainable food system in the state of Rio Grande do Sul, Brazil].What is the research aim or objective or research questions?

We changed our abstract.

2.       Your discussion of food systems is idealistic, and the use of Van Braun's two criteria are open to question and debate. All food systems ' would claim to be 'suitable for the purpose in question': dominant global food chains would claim this on the basis of efficiency and the purpose of profit, which is their primary purpose. No 2 is really unclear to me, 'they must define domains for policy and programmatic priorities so as  not to exclude all economic, social, and ecological dimensions of sustainability' so if it does not address this then it is not a food system? None of the above addresses equity, food security or access.

Here, we are speaking about sustainable food systems. Even though we recognize the importance of other kinds of food system, and this take part of another debates. When we speak about PNAE we focus on sustainable food systems that started from act 11,947. Before that, the food used in PNAE were from huge food chains, and this did not work in Brazil. We had problems in logistics and, that time, the main part of products consumed were industrialized. With introduction of act 11,947 was possible guarantee food security, so our debate also englobes the importance of act 11,947 that made the transition to a traditional food system to one more sustainable.

3.      The introduction and the section on page 2 is good but is there a need for the later section on  theoretical framework and should this be labelled epistemology? More detailed comments on this below.

Social learning theory and methods need to be addressed more clearly in the final discussion of results. Think of Freire and pedagogy of the oppressed. I am not clear from the results how group or organisation learning takes place [ see page 2 where you say ' Based on this context, the question arises: How can learning influence the construction of a food system? ']. Is this a research question? How does this relate to the research aim or objectives?

It’s a good point a section about epistemology, but this item we would like to use in another article from this research. This article is too big for that.

We hope to have improved our results explained better this transition from individual learning to Political Economy. You can observe this change on page 8, lines 377-385. And on 4.8 item on page 15 (lines 753-827), besides item 5 (lines 831-886).

4.      Section 2, and 21. needs to be abbreviated. They is much too long and too much detail. There is repetition in the section from earlier on eg von Braun.

The detail on political economy approaches is not reflected in the methodology, what is its purpose here?  I recommend your seriously cut this section down to a paragraph or two not the existing page and a half it currently occupies.

We shortened this section to 8 paragraphs (see on page 3).

We also improved our methodology to make clear Political Economy approaches (see on pages 9 and 10).

5.      Section 2.2 is important but can still be shortened, I think that you need to address how organisations learn and indeed don't learn. As it stands this is a section which deals with 'could' dos, it is not clear how it applies to the current research. The focus is largely on individual learning not organisational learning?

We shortened a little, still keeping 9 paragraphs. The focus here is explain individual learning to collective learning. But to understand collective learning, its necessary know the individual learning because a collective learning is a result of learnings from each person that make part of organization.

6.      Section 2.3 what happened between the 1960s and the 2000s? How did the Bolsanaro reforms impact on the programme?

This is an important section and could be re-introduced in the introduction as a policy context.

I was not clear if universities are included in the scope of the program [page 7, line 350].

We shortened this section, because we worked this period (1960-2000) in another article published by us, so after another reviewer's observation, we decide cut this historic.

About Bolsonaro reforms there is not to say. At the beginning of his

mandate he wanted to reduce FNDE powers, but this not go ahead because pandemic period. So he kept the program and gave more autonomy to each municipality solve their problems. The one thing that he did was to increase the limit of values to agricultures sell to program (from 20 thousand reais to 40 thousand reais). But he did not change the values paid for students meals.

PNAE is destinated to basic education, i.e., from kindergarten to high school (see on lines 264-265 on page 6). Sometimes attends young people and adults that do not finished basic education. In Brazil, people that lives in vulnerability are the focus of PNAE, mainly children that don’t eat enough to improve their cognition. People that reach universities passed from this process and there is no need a specific program. There are universities that have similar programs as PNAE, but this is not topic to this article.

7.      Section 2.4 analysis  framework how was this arrived at, was this the 'framework to analyze learnings'?  it is not clear to me how you arrived at this and was it piloted or validated before using it? I don't see the relationship between political economy and social learning, please expand on this.

See lines 377-390. This a theoretical framework, not a model. We used critical approaches in a qualitative study. In an abductive logic, after data collection we realized that the approach of Political Economy could work with learning levels.

8.      Page 11, line 481  under section 3 you say 'In order to achieve the proposed objective, a qualitative-descriptive research was selected [19, 20, 21], with abductive logic [15, 22, 23], using a multiple-case study [24, 25]  and semi-structured interviews [26] as a strategy', so what objective? So is this a case study using data from interviews and existing reports?  Is it a comparative case study approach?  it seems like you have in effect what Yin refers to as 'nested cases' compiled  using a range of methods including semi-structured interviews with a range of stakeholders, document analysis, geographical mapping, as well as measurements or indicators of the food environment.Case studies are comprised of two parts including a subject, which forms the focus of the research, and an analytical frame, which provides a means of interpreting it or placing in into a context (Thomas, 2015).

We liked your suggestion about the book “The Anatomy of the Case Study”

by Gary Thomas e Kevin Myers (2015), this made work more in explain about case study using Yin (2018) (see on page 9, 408-443 lines). And this made us create a table to compare al municipalities visited (see on page 16)

9.      Abductive logic is a tool of analysis not a method of research? 

Table 1 is not needed in the main document and could be submitted as supplementary material, we don't need the interviewee code or date of interview. Please put in a generic description such an interviews were held between 'date' and 'date' and clarify if any of this took place during COVID lockdowns or restrictions with a range of interviewees from X to Y.

We moved Table 1 to Appendix. We did not work during lockdowns. If you notice our first interview occurred only on October 2021 and the main part of interviews happened on Jully 2022. In this period was not restrictions in Brazil.

10.   So you now present 7 case studies which are fine but seem to come an be complied from a range of sources including documentary sources, interviews etc. No interviewee data is presented suggesting that these were subservient to the purpose of developing case studies and are not being used to provide comparison across these data sources eg interviews

Perhaps a table pulling together the similarities and differences between the 7 case studies would help the reader.

We accepted your suggestion, so we create a table on page 16 to compare all cases.

11.   What are you adding to the research agenda, what is unusual or new about your findings?

 We talked about it on conclusion, page 21 (890-899 lines).

12.   You have another table 1 on page 24 called 'Relationship between learning levels and Political Economy' which I do not understand, the narrative does not help me much so what does the following mean ' In regions where learning is communicative, there is a concern...'. What does communicative mean in this context of organisation or regions?

Hope that is better understandable after 4.8 item on page 15 (lines 753-827), and item 5 (lines 831-886).

13.   There needs to be a statement of ethical approaches and approval, at the very least and indication of good research governance is required.

We were approved by Term 19/2551-0001809-6 in Research Support Foundation of the State of Rio Grande do Sul (Fundação de Amparo à Pesquisa do Estado do Rio Grande do Sul – FAPERGS).

14.   Can you check how many times you use 'therefore' in most cases you don't need to use it just remove the word.

We reduced to 4 times.

Reviewer 2 Report

The National School Feeding Program is one of the oldest public policies in Brazil, but its conception has changed  over the years. The PNAE was influenced, in the 1960s, by the foods that made up the programs established by the United States Agency for International Development (USAID) and the World  Food Program (WFP) of the United Nations.  This article sought to reconcile approaches on food systems and Political Economy with levels of learning, as it is believed that the actions of leaders who dictate the direction of a food system have their origin in how such individuals learn and transmit their  knowledge. By understanding how knowledge construction occurs, it is possible to ensure  a better understanding of political decisions, which consequently influence the economy.  During the 1970s and 1990s, processed foods  such as cookies, chocolate and preserves were the basis of school meals. It was only in 27 2009, with the introduction of Act 11,947 , that it was determined that at least 30% of 928 food should come from family farming. This measure introduced fresh and healthy food  in schools, stimulating the local economy.  What really changed was the understanding of program managers from  the FNDE to the municipalities that receive their resources. Paradigm shifts such as this  are due to changes in levels of learning. Therefore, before making political and economic  assessments of a food system, it is necessary to assess: Who learns?; Why do they learn?; How do they learn? and What do they learn?. Thus, it will be more appropriate and responsible to reflect on the answers to the questions: Who owns what?; Who does what?;  Who gets what?; and What do they do with it?. As a positive point, this research reinforced the importance of the PNAE, both in the  nutritional aspect (which favors the students’ cognition, in addition to benefiting their  health through the ingestion of natural and fresh foods, without the use of industrial additives), and in the implication of an institutional market (which promotes access to income for family farmers, as well as local development). It should be noted that the research contributes to helping family farmers understand their place in the world and, consequently, their importance in the food production process. By reaffirming their position, they may fight for improvements in public policy that can reflect and improve their reality. The main difficulty of this research was the limitation of official data on family farming in the PNAE, which had not been updated since 2017. Another limiting factor was the  Coronavirus pandemic, which disrupted sectoral councils, as well as relations between  the Education Secretariats with groups of farmers.

The paper is well done but I have some remarks, 

- Some graphic charts of the statistical data could improve the paper quality 

- Moderate editing of English language is required

Moderate editing of English language is required

Author Response

Dear Reviewer 2

We appreciated your comments and suggestions. They contributed for some corrections and changes in our article.

We would like to answer your points showing the main changes that we done.

Best regards

1.      The selection of stakeholders is not clear. The characteristics of the interviewed people is reported in the Table 1, but the authors should explain the criteria that had lead this choice.

We also improved our methodology to make it clear (see on pages 9 and 10).

2.      Some information (p.e., the date of the interviews or the total time spent for the survey) are irrilevant for the aims of the study.

We moved Table 1 to Appendix.

3.      The results need to be graphically represented. It is really difficult reading the findings without the support of a representation.

We accepted your suggestion, so we create a table on page 16 to compare all cases.

We hope to have improved our results explained better this transition from individual learning to Political Economy. You can observe this change on page 8, lines 377-385. And on 4.8 item on page 15 (lines 753-827), besides item 5 (lines 831-886).

Reviewer 3 Report

This is potentially very interesting and is the basis of a very interesting paper but it needs restructuring and some parts to be rewritten. Comments follow.

The abstract would lead me too think this is about the development of a tool for gathering data [Therefore, the purpose of this article was to prepare a framework to analyze learnings ], but later on various terms are used with a confusing effect, so aim, purpose etc, see page 9 [The aim of this article is to analyze the lessons learned in the PNAE that contribute to the development of a sustainable food system in the state of Rio Grande do Sul, Brazil].What is the research aim or objective or research questions? Please clarify and be consistent throughout article. On page 3 line 98+ you say 'Therefore, the goal of this article was to prepare a framework to analyze the learnings in  the PNAE, which contribute to the development of a sustainable food system, in the state  of Rio Grande do Sul.' And on page 9 you say 'The aim of this article is to analyze the lessons learned in the PNAE that contribute  to the development of a sustainable food system in the state of Rio Grande do Sul, Brazil.  Therefore, with regard to creating an analysis framework, the three perspectives – food  systems, political economy, and learning – are useful to ensure the partial achievement of  this goal.' So is it a goal or an aim and what is the objective?

Your discussion of food systems is idealistic, and the use of Van Braun's two criteria are open to question and debate. All food systems ' would claim to be 'suitable for the purpose in question': dominant global food chains would claim this on the basis of efficiency and the purpose of profit, which is their primary purpose. No 2 is really unclear to me, 'they must define domains for policy and programmatic priorities so as  not to exclude all economic, social, and ecological dimensions of sustainability' so if it does not address this then it is not a food system? None of the above addresses equity, food security or access. 

In many ways the reality is that public sector initiatives such as described in this article are the result of failures in the dominant food system.

The introduction and the section on page 2 is good but is there a need for the later section on  theoretical framework and should this be labelled epistemology? More detailed comments on this below. 

Social learning theory and methods need to be addressed more clearly in the final discussion of results. Think of Freire and pedagogy of the oppressed. I am not clear from the results how group or organisation learning takes place [ see page 2 where you say ' Based on this context, the question arises: How can learning influence the construction of a food system? ']. Is this a research question?How does this relate to the research aim or objectives? 

Section 2, and 21. needs to be abbreviated. They is much too long and too much detail. There is repetition in the section from earlier on eg von Braun. 

The detail on political economy approaches is not reflected in the methodology, what is its purpose here?  I recommend your seriously cut this section down to a paragraph or two not the existing page and a half it currently occupies. 

Section 2.2 is important but can still be shortened, I think that you need to address how organisations learn and indeed don't learn. As it stands this is a section which deals with 'could' dos, it is not clear how it applies to the current research. The focus is largely on individual learning not organisational learning?

Section 2.3 what happened between the 1960s and the 2000s? How did the Bolsanaro reforms impact on the programme?

This is an important section and could be re-introduced in the introduction as a policy context. 

I was not clear if universities are included in the scope of the program [page 7, line 350]. 

Section 2.4 analysis  framework how was this arrived at, was this the 'framework to analyze learnings'?  it is not clear to me how you arrived at this and was it piloted or validated before using it? I don't see the relationship between political economy and social learning, please expand on this.

Page 11, line 481  under section 3 you say 'In order to achieve the proposed objective, a qualitative-descriptive research was selected [19, 20, 21], with abductive logic [15, 22, 23], using a multiple-case study [24, 25]  and semi-structured interviews [26] as a strategy', so what objective? So is this a case study using data from interviews and existing reports?  Is it a comparative case study approach?  it seems like you have in effect what Yin refers to as 'nested cases' compiled  using a range of methods including semi-structured interviews with a range of stakeholders, document analysis, geographical mapping, as well as measurements or indicators of the food environment.Case studies are comprised of two parts including a subject, which forms the focus of the research, and an analytical frame, which provides a means of interpreting it or placing in into a context (Thomas, 2015). 

Abductive logic is a tool of analysis not a method of research?  

Table 1 is not needed in the main document and could be submitted as supplementary material, we don't need the interviewee code or date of interview. Please put in a generic description such an interviews were held between 'date' and 'date' and clarify if any of this took place during COVID lockdowns or restrictions with a range of interviewees from X to Y. 

My personal presence is for a section called findings not resultless especially for qualitative data, but happy to bow to journal requirements.

So you now present 7 case studies which are fine but seem to come an be complied from a range of sources including documentary sources, interviews etc. No interviewee data is presented suggesting that these were subservient to the purpose of developing case studies and are not being used to provide comparison across these data sources eg interviews 

Perhaps a table pulling together the similarities and differences between the 7 case studies would help the reader. 

Your discussion as it currently stands is a repeat or reiteration of the findings, it needs to be a critical discussion vis a vis exiting literature.  So we found this and others such as X  agree/disagree with this. 

What are you adding to the research agenda, what is unusual or new about your findings?

You have another table 1 on page 24 called 'Relationship between learning levels and Political Economy' which I do not understand, the narrative does not help me much so what does the following mean ' In regions where learning is communicative, there is a concern...'. What does communicative mean in this context of organisation or regions?

There needs to be a statement of ethical approaches and approval, at the very least and indication of good research governance is required. 

This is an important article detailing the workings of the PNAE system, how it works and of great interest to many outside Brazil who look to programs such as this for inspiration. 

Some minor issues to be addressed

Can you check how many times you use 'therefore' in most cases you don't need to use it just remove the word.

Not so much English grammar or syntax but there is a lot of repetition eg the following appears at least twice if not 3 times: 'The state of Rio Grande do Sul, for example, has an agricultural production that includes crops such as soy, corn, wheat, rice, beans, tobacco, grapes, apples, citrus, and olive  trees. This state has 11,088,065 inhabitants [17], of which 2.4 million are students [18]. It comprises 497 municipalities divided into seven mesoregions – Northeast, Northwest, 95 Southeast, Southwest, West Central, East Central, and Metropolitan Area [17]. In each city,' in each mesoregion, it is possible to identify the diversity in the execution of the PNAE'. 

Author Response

Dear Reviewer 3

We appreciated your comments and suggestions. They contributed for some corrections and changes in our article.

We would like to answer your points showing the main changes that we done.

Best regards

1.      The Abstract has to be reconsidered, by not using abbreviations. Also, there are missing the main results and their implications. The introductory part should be drastically reduced.

Among the keywords, the authors are asked to use suggestive keywords and to avoid general ones, like learning; Political Economy, etc. For better visibility on databases, the authors are asked not to repeat among keywords the words/concepts included in the title of the article. Entering different words in the title and in the keywords can improve the search for the paper in metasearch engines and internet databases.

We changed our abstract, but we decided to keep the abbreviations because PNAE (National School Feeding Program) is recognized by this abbreviation.

We decided to keep the keywords.

2.       In the introduction, the presentation of the structure of the paper is missing. Anyway, the objective of the manuscript is clearly stated.

Ok, we decided to keep.

3.      The part of the 2. Theoretical Framework is far too long and it soi presenting unnecessary information, irrelevant for the needs to construct the literature gap by presenting the focus of the current study. It is advisable to condense it and to make it clearer and easier to follow.

We shortened this section.

We also shortened the section 2.3, because we worked this period (1960-2000) in another article published by us, so after another reviewer's observation, we decide cut this historic.

4.      The methodology part is not well conceptualized. For instance, there is need to present how the sample was constructed and why the it is representative for the entire studied population (presenting similar studies would help validate the method).

Table 1 is to long and difficult to follow, I suggest to try to make it shorter (and, additionally, to move the entire table as Annex at the end of the article)

We work more in explain about case study using Yin (2018) (see on page 9, 408-443 lines). And this made us create a table to compare al municipalities visited (see on page 16)

We also improved our methodology to make clear political economy approaches (see on pages 9 and 10).

We moved Table 1 to Appendix.

5.      The results. That part is missing, is failing in presenting the results of the study. Presentation of the 7 municipalities is not result of your study, but the part of studied material (so it should be moved to that part). The presentation must be condensed, presented as a comparative analysis (using tables, figures), to reduced it at the minimum. As it is, seems to be too long and futile.

Hope that is better understandable after 4.8 item on page 15 (lines 753-827), besides item 5 (lines 831-886). We also create a table to compare al municipalities visited (see on page 16)

6.      The discussion. The authors confuse results with discussions, they are mess up on the chapter 5. Discussion on Learnings. For instance, Table 2 seems to be part of results. The authors fail in presenting an adequate comparison of the results with the previous literature; so the authors must emphasize the contribution of the manuscript to the literature, leading to theoretical implications.

We talked about contribution of the manuscript on conclusion page 21 (890-899 lines).

We also rewrite the results (partially) and the conclusion.

Reviewer 4 Report

The paper is focused on suggesting a framework to analyze learnings in
the National School Feeding Program - aimed to contribute to the development of a sustainable food system - in a state of Brazil. Based on 8 research questions, the study is original and relevant, annd the methodological approach is suitable.
However, I suggest some adjustments in order to improve the quality of the paper:

1. The selection of stakeholders is not clear. The characteristics of the interviewed people is reported in the Table 1, but the authors should explain the criteria that had lead this choice.

2. Some information (p.e., the date of the interviews or the total time spent for the survey) are irrilevant for the aims of the study.

3. The results need to be graphically represented. It is really difficult reading the findings without the support of a representation.

Author Response

Dear Reviewer 4

We appreciated your comments and suggestions. They contributed for some corrections and changes in our article.

We would like to answer your points showing the main changes that we done.

Best regards

1.      The selection of stakeholders is not clear. The characteristics of the interviewed people is reported in the Table 1, but the authors should explain the criteria that had lead this choice.

We also improved our methodology to make it clear (see on pages 9 and 10).

2.      Some information (p.e., the date of the interviews or the total time spent for the survey) are irrilevant for the aims of the study.

We moved Table 1 to Appendix.

3.      The results need to be graphically represented. It is really difficult reading the findings without the support of a representation.

We accepted your suggestion, so we create a table on page 16 to compare all cases.

We hope to have improved our results explained better this transition from individual learning to Political Economy. You can observe this change on page 8, lines 377-385. And on 4.8 item on page 15 (lines 753-827), besides item 5 (lines 831-886).

Reviewer 5 Report

Dear Authors,

your paper entitled “National School Feeding Program (PNAE): A public policy that promotes a learning framework and a more sustainable food system, in Rio Grande do Sul, Brazil” is interesting and useful to understand some of the dynamics of sustainable food system in Brazil and the relationship between family farming, school meals and food supply.

The authors set out to identify a framework for analyzing PNAE learnings, which contribute to the development of a sustainable food system. To this end, the authors conducted a qualitative-descriptive survey, with abductive logic, using a study of multiple cases and semi-structured interviews.

From an overall point of view, the contribution is well written and explained, the structure is good and the contents are quite interesting.

Some considerations follow:

The “Abstract” section is clear but, in my humble opinion, 8 questions to show the objectives are not the best solution. Furthermore, I would suggest adding some indications on the results and implications.

The "Introduction" section is clear and provides the necessary information to understand the structure of the paper and the objectives of the study.

The “Theoretical framework” section is clear but I think it is too long and, furthermore, several parts are similar to the sources used, including 'The PNAE (National School Feeding Program) activity system and its mediations', published in Frontiers of environmental science.

The "Methodology" section is clear, well explained and provides information that makes it easy to understand what was done and achieved. I suggest moving Table 1 to the Appendix to make the document more readable and, given the importance due to the length of the interviews, it might be useful to insert a new column indicating the timing of each interview. Furthermore, I would suggest introducing a sub-section dedicated to the presentation of the "study area" to focus on the geographical aspect of the study.

The “Results” and “Discussion on learning” sections are clear and well explained.

The 'Conclusions' section is clear and understandable and highlights the main aspects that emerged from the study but, at the same time, does not stimulate international interest. In cases like yours, I think it is very useful to highlight how a local study can be of interest to international researchers. I therefore suggest emphasizing further the description of the implications of the results obtained not only in terms of local improvement but also in a global view.

Lines 987-1028 are not formatted.

In conclusion, I appreciate the topic of the paper and consider your study interesting and rather original; therefore, in my humble opinion, I think it can be published in the IF journal as Foods, after a thorough review.

Author Response

Dear Reviewer 5

We appreciated your comments and suggestions. They contributed for some corrections and changes in our article.

We would like to answer your points showing the main changes that we done.

Best regards

1.      From an overall point of view, the contribution is well written and explained, the structure is good and the contents are quite interesting.

Ok. Thank you!

2.       The “Abstract” section is clear but, in my humble opinion, 8 questions to show the objectives are not the best solution. Furthermore, I would suggest adding some indications on the results and implications.

We changed our abstract in accord with your suggestions.

3.      The "Introduction" section is clear and provides the necessary information to understand the structure of the paper and the objectives of the study.

Ok. Thank you!

4.      The “Theoretical framework” section is clear but I think it is too long and, furthermore, several parts are similar to the sources used, including 'The PNAE (National School Feeding Program) activity system and its mediations', published in Frontiers of environmental science.section down to a paragraph or two not the existing page and a half it currently occupies.

We shortened this section and agree with your observation about the article published in Frontiers of Environmental Science.

5.      The "Methodology" section is clear, well explained and provides information that makes it easy to understand what was done and achieved. I suggest moving Table 1 to the Appendix to make the document more readable and, given the importance due to the length of the interviews, it might be useful to insert a new column indicating the timing of each interview. Furthermore, I would suggest introducing a sub-section dedicated to the presentation of the "study area" to focus on the geographical aspect of the study.

We work more in explain about case study using Yin (2018) (see on page 9, 408-443 lines). And this made us create a table to compare al municipalities visited (see on page 16).

We also moved Table 1 to Appendix.

6.      The 'Conclusions' section is clear and understandable and highlights the main aspects that emerged from the study but, at the same time, does not stimulate international interest. In cases like yours, I think it is very useful to highlight how a local study can be of interest to international researchers. I therefore suggest emphasizing further the description of the implications of the results obtained not only in terms of local improvement but also in a global view.

We talked about it on conclusion, page 21 (890-899 lines).

Hope that is better understandable after 4.8 item on page 15 (lines 753-827), and item 5 (lines 831-886).

Reviewer 6 Report

The topic addressed is relevant for the journal and for the field of study, in general. This study aims to prepare a framework to analyze learnings in the Brazilian National School Feeding Program, which contribute to the development of a sustainable food system, in the state of Rio Grande do Sul, Brazil.

The subject area is rather interesting, and, possibly, not enough approached by other scholars, so there is potential room for this manuscript to bring new information, once it reaches the expected level of quality.   

The Abstract has to be reconsidered, by not using abbreviations. Also, there are missing the main results and their implications. The introductory part should be drastically reduced.

Among the keywords, the authors are asked to use suggestive keywords and to avoid general ones, like learning; Political Economy, etc. For better visibility on databases, the authors are asked not to repeat among keywords the words/concepts included in the title of the article. Entering different words in the title and in the keywords can improve the search for the paper in metasearch engines and internet databases.

In the introduction, the presentation of the structure of the paper is missing. Anyway, the objective of the manuscript is clearly stated.

The part of the 2. Theoretical Framework is far too long and it soi presenting unnecessary information, irrelevant for the needs to construct the literature gap by presenting the focus of the current study. It is advisable to condense it and to make it clearer and easier to follow.

The methodology part is not well conceptualized. For instance, there is need to present how the sample was constructed and why the it is representative for the entire studied population (presenting similar studies would help validate the method).

-        Table 1 is to long and difficult to follow, I suggest to try to make it shorter (and, additionally, to move the entire table as Annex at the end of the article)

The results. That part is missing, is failing in presenting the results of the study. Presentation of the 7 municipalities is not result of your study, but the part of studied material (so it should be moved to that part). The presentation must be condensed, presented as a comparative analysis (using tables, figures), to reduced it at the minimum. As it is, seems to be too long and futile.

The discussion. The authors confuse results with discussions, they are mess up on the chapter 5. Discussion on Learnings. For instance, Table 2 seems to be part of results. The authors fail in presenting an adequate comparison of the results with the previous literature; so the authors must emphasize the contribution of the manuscript to the literature, leading to theoretical implications.

The conclusions’ part is too long, also, being dedicated more to present the results, 

Author Response

Dear Reviewer 6

We appreciated your comments and suggestions. They contributed for some corrections and changes in our article.

We would like to answer your points showing the main changes that we done.

Best regards

1.      The Abstract has to be reconsidered, by not using abbreviations. Also, there are missing the main results and their implications. The introductory part should be drastically reduced.

Among the keywords, the authors are asked to use suggestive keywords and to avoid general ones, like learning; Political Economy, etc. For better visibility on databases, the authors are asked not to repeat among keywords the words/concepts included in the title of the article. Entering different words in the title and in the keywords can improve the search for the paper in metasearch engines and internet databases.

We changed our abstract, but we decided to keep the abbreviations because PNAE (National School Feeding Program) is recognized by this abbreviation.

We decided to keep the keywords.

2.       In the introduction, the presentation of the structure of the paper is missing. Anyway, the objective of the manuscript is clearly stated.

Ok, we decided to keep.

3.      The part of the 2. Theoretical Framework is far too long and it soi presenting unnecessary information, irrelevant for the needs to construct the literature gap by presenting the focus of the current study. It is advisable to condense it and to make it clearer and easier to follow.

We shortened this section.

We also shortened the section 2.3, because we worked this period (1960-2000) in another article published by us, so after another reviewer's observation, we decide cut this historic.

4.      The methodology part is not well conceptualized. For instance, there is need to present how the sample was constructed and why the it is representative for the entire studied population (presenting similar studies would help validate the method).

Table 1 is to long and difficult to follow, I suggest to try to make it shorter (and, additionally, to move the entire table as Annex at the end of the article)

We work more in explain about case study using Yin (2018) (see on page 9, 408-443 lines). And this made us create a table to compare al municipalities visited (see on page 16)

We also improved our methodology to make clear political economy approaches (see on pages 9 and 10).

We moved Table 1 to Appendix.

5.      The results. That part is missing, is failing in presenting the results of the study. Presentation of the 7 municipalities is not result of your study, but the part of studied material (so it should be moved to that part). The presentation must be condensed, presented as a comparative analysis (using tables, figures), to reduced it at the minimum. As it is, seems to be too long and futile.

Hope that is better understandable after 4.8 item on page 15 (lines 753-827), besides item 5 (lines 831-886). We also create a table to compare al municipalities visited (see on page 16)

6.      The discussion. The authors confuse results with discussions, they are mess up on the chapter 5. Discussion on Learnings. For instance, Table 2 seems to be part of results. The authors fail in presenting an adequate comparison of the results with the previous literature; so the authors must emphasize the contribution of the manuscript to the literature, leading to theoretical implications.

We talked about contribution of the manuscript on conclusion page 21 (890-899 lines).

We also rewrite the results (partially) and the conclusion.

Round 2

Reviewer 5 Report

Dear Authors,

I appreciate the changes and additions made that I requested. I think the current version is clear, well structured and in line with the purpose of the journal. I therefore believe it is suitable for publication.

Best regards

Reviewer 6 Report

Dear authors,

For quality and visibility reasons, I do not share your opinion on not changing the keywords and using abbreviations in the Abstract. But I will let you decide that.

The manuscript was, clearly improved in the second submission.